# LncRNA PNKY Is Upregulated in Breast Cancer and Promotes Cell Proliferation and EMT in Breast Cancer Cells

**DOI:** 10.3390/ncrna9020025

**Published:** 2023-04-06

**Authors:** Forough Hakiminia, Firooz Jannat Alipoor, Mostafa Keshavarz, Malek Hossein Asadi

**Affiliations:** Department of Biotechnology, Institute of Science and High Technology and Environmental Sciences, Graduate University of Advanced Technology, Kerman 7631885356, Iran

**Keywords:** PNKY, cancer stem cells, breast cancer, EMT, cell senescence

## Abstract

Long non-coding RNAs (lncRNAs) are known to be important regulators in different cellular processes and are implicated in various human diseases. Recently, lncRNA PNKY has been found to be involved in pluripotency and differentiation of embryonic and postnatal neural stem cells (NSCs); however, its expression and function in cancer cells is still unclear. In the present study, we observed the expression of PNKY in various cancer tissues, including brain, breast, colorectal, and prostate cancers. In particular, we demonstrated that lncRNA PNKY was significantly upregulated in breast tumors, especially high-grade tumors. Knock down experiments indicated that the suppression of PNKY in breast cancer cells could restrict their proliferation by promoting apoptosis, senescence, and cell cycle disruption. Moreover, the results demonstrated that PNKY may play a crucial role in the cell migration of breast cancer cells. We further found that PNKY may trigger EMT in breast cancer cells by upregulating miR-150 and restricting the expression of Zeb1 and Snail. This study is the first to provide new evidence on the expression and biological function of PNKY in cancer cells and its potential contribution to tumor growth and metastasis.

## 1. Introduction

Long non-coding RNAs are identified as a type of non-coding RNA that is longer than 200 nucleotides, and which does not have the potential to be translated into protein. Long non-coding RNAs influence various cellular regulatory processes, including chromatin remodeling, RNA processing, transcriptional regulation, and translational control. Recent evidences suggest that the dysregulation of lncRNAs could contribute to vital aspects of cancer biology, including tumorigenesis, tumor progression, and metastasis [1,2,3]. Recent studies have also indicated that some lncRNAs play a central role in both embryonic stem cells and cancer cells [4]. For instance, lncRNA MIAT, ES3, and ES1 not only contribute to maintaining pluripotency in embryonic stem cells, but they also play a crucial role in tumorigenesis, tumor progression, and metastasis [5,6,7,8]. PNKY is also a long non-coding RNA that regulates the pluripotency and differentiation of embryonic and postnatal neural stem cells (NSCs). It is 1560 nucleotides long, located on chromosome 6q16.1, and been determined as an evolutionary conserved polyadenylated transcript. Recently, PNKY has been found to be involved in mRNA splicing through interactions with polypyrimidine tract-binding protein 1 (PTBP1), a well-known regulator of mRNA splicing, during the neuronal differentiation of NSCs [9]. However, the function of the PNKY transcript has yet to be defined and there are no studies on the exact activity of PNKY in cancer tissues and cells, hence our interest in examining its expression and biological role in human breast tumors.

Thus, we investigated the potential expression of lncRNA PNKY in a series of diverse human cancer RNA pools extracted from ten tumor tissue samples, including breast, brain, colorectal, and prostate tumors, as well as a series of stem/cancer cells by RT-qPCR. By conducting a detailed examination, we assessed the potential expression of PNKY transcripts in a series of human breast tumor tissues and their potential biological function in two breast cancer cell lines, MDA-MB-231 and MCF7, using RNAi technology. 

## 2. Results

### 2.1. PNKY Is Expressed in Different Cancer Types and Its Expression Is Strongly Upregulated in High-Grade Breast Cancer

In the first phase of our study, we measured the expression of PNKY in various cancer tissues. The PNKY transcript consists of one exon; therefore, for each sample, a no-RT control was used in parallel to detect any potential contamination with genomic DNA. In order to investigate the potential expression of PNKY in colorectal, brain, breast, and prostate cancers, we detected the PNKY expression in the RNA pool containing ten RNAs extracted from cancer tissue samples of each type of cancer, including breast, brain, colorectal, and prostate cancer (Figure 1A). The preliminary results revealed that the PNKY transcript is expressed in breast, brain, prostate, and colorectal cancer tissues. This data revealed for the first time that the PNKY transcript is expressed in various cancer tissues. It is proposed that the non-coding transcript may play an important role in cancer cell biology. In particular, we demonstrated PNKY’s expression in a series of breast cancer tissues and cell lines. The RT-qPCR analysis data revealed that the expression level of PNKY is significantly higher in breast tumor tissues than in their paired non-tumor tissues from the same donor (Table 1, *p* = 0.04). Additionally, we observed the significant upregulation of PNKY in high-grade breast tumors versus low-grade tumors (Figure 1B, *p* < 0.01). We further found that PNKY was downregulated in lobular breast tumor samples compared with ductal breast tumor tissues, but it was broadly significant (Table 1, *p* = 0.07). Furthermore, the gene expression results demonstrated that PNKY was significantly upregulated in positive lymph node invasive breast tumors compared with negative ones (Table 1, *p* = 0.011). Our data indicate an association between PNKY’s expression and the clinicopathological properties of breast tumors, such as HER2, P53, ER, and PR status. The results revealed that PNKY is downregulated in ER^+^, PR^+^, and Her2^+^ tumor tissues compared with ER^−^, PR^−^, and Her2^−^ tumor samples, respectively (Figure 1C). Additionally, the data from RT-qPCR analysis indicated that the expression of PNKY was apparently higher in P53 negative breast tumors compared with P53 positive tumor samples; however, the difference was not statistically significant (Table 1, *p* = 0.17). These results suggest that PNKY upregulation is related to breast tumor development and progression.

### 2.2. PNKY Transcript Was Localized in the Nucleus and Its Expression Was Significantly Downregulated during Neural Differentiation 

In the next step of our study, we investigated the expression of PNKY in a series of stem and cancer cells using RT-PCR. We observed the expression of PNKY in NCCIT (embryonic carcinoma cell line), dental pulp stem cells (adult stem cells), as well as MDA-MB-231 and MCF7 (breast cancer cell lines) (Figure 2A). We next found that the PNKY transcript was localized in the nucleus of MDA-MB-231 cancer cells by using subcellular fractionation assay (Figure 2B, *p* < 0.01). To verify the role of PNKY in differentiation, we evaluated PNKY expression during the neural differentiation of NCCIT cells. Our data showed that the expression level of PNKY, as well as stemness factors such as OCT4, Nucleostemin, and NANOG, were significantly downregulated during neural differentiation (Appendix A). 

### 2.3. PNKY Promotes Cell Cycle Progression and Induces Proliferation of Breast Cancer Cells

To explore the role of PNKY in breast cancer cells, we first repressed PNKY expression in MDA-MB-231 and MCF7 using specific PNKY siRNAs. Then, we investigated the influence of this suppression on proliferation and cell cycle progression. Two days post-transfection, RT-qPCR data established that the expression of PNKY was efficiently suppressed by the means of specific PNKY siRNAs (Figure 2C). The cell proliferation results indicated that the downregulation of PNKY significantly restricted breast cancer cell proliferation (Figure 2D). Additionally, a colony formation assay demonstrated that si-PNKY transfected cells were less able to form colonies compared with the cells that were transfected with scrambled siRNA (Figure 2E). Furthermore, flowcytometry results revealed that PNKY suppression elevated the proportion of cancer cells in the G1 phase as well as decreasing the percentage of cells in the S phase of the cell cycle (Figure 3A). 

In line with the MTT data and cell cycle analysis, the well-known genes that regulate cell proliferation and cell cycle progression, including Cyclin D1 and PCNA, were significantly downregulated when PNKY was knocked down in breast cancer cells (Figure 3B). Altogether, these results suggest that PNKY may contribute to the proliferation of cancer cells through cell cycle regulation.

### 2.4. PNKY Induces Migration and Suppresses Apoptosis in MDA-MB-231 and MCF7 Cells

The overexpression of PNKY in high-grade and lymph node metastasis breast tumors indicates that the PNKY transcript may play a crucial role in breast tumor progression. To determine this, we measured and monitored cell migration inward to fill the scratch-wound void in MDA-MB-231 and MCF7 cells with or without PNKY suppression. The results showed that the migration of MDA-MB-231 and MCF7 cells was significantly reduced in PNKY knocked down cells compared with the cells that were transfected with scrambled siRNA (Figure 4A). Apoptosis is a well-defined mechanism that regulates the hemostasis of normal tissues, and its disruption leads to tumor initiation and progression. Thus, we evaluated the impact of PNKY silencing on apoptosis and measured the proportion of apoptotic cells with or without PNKY downregulation. We achieved this by staining the cells with Annexin-V and propidium iodide and analyzing them using flowcytometry. As expected, the downregulation of PNKY induces apoptosis in MDA-MB-231 and MCF7 cells and anticipatedly alters the expression level of the well-known apoptosis markers, Bax and Bcl2 (Figure 4B,C). These data suggest that PNKY may be an onco-lncRNA and anti-apoptotic factor that may contribute to the development and progression of breast cancer. 

### 2.5. PNKY Restricts Cellular Senescence in MDA-MB-231 and MCF7 Cells

Many studies have demonstrated that the induction of cellular senescence, like apoptosis, is known as a barrier that limits tumor progression. siRNA-mediated depletion of PNKY in MDA-MB-231 and MCF7 cells suppressed cell migration, leading to G1 arrest and apoptosis induction. Therefore, we speculated that PNKY may play an important role in the senescence process. To verify this, we investigated the impact of PNKY suppression on cellular senescence by detecting the senescence-associated β-galactosidase activity following PNKY inhibition in breast cancer cells. The results revealed that the proportion of senescent cells was significantly raised in PNKY knocked down cells compared with cells treated with scrambled siRNA (Figure 5A). Consistent with the above data, the expression of well-known senescence markers, p16^Ink4A^ and Cox2, was significantly upregulated when PNKY was suppressed (Figure 5B). Collectively, the emerging data indicate that PNKY can contribute to breast tumor cell proliferation by inhibiting apoptosis and cellular senescence.

### 2.6. PNKY Induces EMT in Breast Cancer Cells

Numerous reports have established that epithelial-to-mesenchymal transition plays a critical role in the progression of cancers by contributing to cancer invasion and metastasis [10,11]. PNKY upregulation in high-grade/stage breast tumors and its impact on cancer cell migration suggests that PNKY may play a fundamental role in tumor metastasis and EMT. Thus, we determined the expression level of well-known EMT markers, including E-cadherin, Snail, Vimentin, and Zeb1, through RT-qPCR and immunofluorescence assays to assess whether PNKY can induce EMT in breast cancer cells. As we anticipated, the immunofluorescence results revealed that E-cadherin upregulation as well as Snail/Vimentin downregulation occurred in breast cancer cells following PNKY silencing. Similarly to the immunofluorescence data, the RT-qPCR results revealed that Zeb1 expression markedly decreased in breast cancer cells when PNKY was suppressed (Figure 6B). Zeb1 is a key transcription factor that controls EMT by regulating the expression of many EMT markers. In addition, the expression of ZEB1 is also regulated by miR-150; therefore, we investigated the expression of miR-150 following PNKY downregulation. Our findings demonstrated that after PNKY suppression in MDA-MB-231 cells, the expression of miR-150 was significantly upregulated, consequently leading to ZEB1 downregulation (Figure 6C). The differential expression of the hsa-miR-29 family has been shown to contribute to EMT. Therefore, we assessed their expression following PNKY downregulation. Our data indicate that the downregulation of PNKY suppressed the hsa-miR-29 family in MDA-MB-231 cells (Figure 6C). Our findings suggest that PNKY may be involved in breast cancer cell metastasis by regulating EMT. 

Our findings revealed that PNKY expression as well as stemness factors were downregulated during neural differentiation. Recent reports have established that the non-coding transcript regulates the pluripotency and differentiation of stem cells. Thus, PNKY can be identified as a stemness factor as it may regulate the expression of stemness transcription factors in cancer cells. Therefore, we investigated the expression of hsa-miR-302, Nanog, and Oct4 expression after PNKY knockdown in breast cancer cell lines. We found that their expression was significantly downregulated when PNKY was silenced (Figure 6B,C). Our findings suggest that PNKY may be a cancer stem cell factor, similarly to Oct4, and that it is involved in regulating the proliferation of CSCs.

## 3. Discussion

Mounting evidence indicates that lncRNAs participate in various cellular regulatory processes, including gene expression, RNA processing, chromatin remodeling, post-transcriptional regulation, and translational control [4]. PNKY is a long non-coding RNA that has been shown to regulate neurogenesis from NSCs in the embryonic and postnatal brain. Recent reports show that PNKY may bind to PTBP1, a well-defined protein known to be a regulator of mRNA splicing during the neuronal differentiation of NSCs [9]. Thus, PNKY is specialized to influence the neuronal differentiation of NSCs through physical interactions with PTBP1 and controls RNA splicing. However, the expression and function of PNKY in cancer cells is largely unknown. Based on cancer stem cell hypotheses, not only coding transcripts such as Oct4, Nucleostemin, Sox2, and Klf4, but also long non-coding transcripts, contribute to regulating CSC proliferation [12]. Therefore, these stemness and reprogramming factors may be involved in tumorigenesis and tumor progression. 

Thus, in the first phase of our study, we profiled the expression of PNKY in several cancer types. The key finding of this study is that PNKY is expressed in different cancer types, including breast, brain, prostate, and colorectal tumors. This result suggests that the PNKY transcript may play an important role in cancer cell biology. In particular, our study is the first to report that PNKY is upregulated in breast tumor tissues and its expression is significantly associated with clinicopathological features of the tumors. Additionally, we demonstrated that the expression of PNKY is markedly upregulated in high-grade breast tumor tissues compared with low-grade ones. This result suggests that the expression of PNKY may be associated with the state of differentiation of cells. It could therefore be used to predict the degree of malignancy of breast tumors. Consistent with the above finding, we identified that the expression of PNKY was noticeably downregulated during neural differentiation. These findings are in accordance with recent reports, which demonstrated that PNKY regulates neuronal differentiation from NSCs. By fractionating the organelles of MDA-MB-231 cells, we found that the human PNKY, similarly to the homologous mouse transcript, is localized in the nucleus. Based on the data, PNKY seems to contribute to gene expression regulation. We further found that PNKY expression is downregulated in lobular breast cancer versus ductal breast cancer. This could be originated from different originating tissues that begin these different breast cancer sub-types. 

To determine the possible causative tumorigenic role of PNKY, we first knocked down its expression by using specific PNKY siRNAs in the MDA-MB-231and MCF7 breast cancer cell lines. The effective inhibition of PNKY was established using RT-qPCR, following which we assessed the impact of its silencing on cellular proliferation and the apoptosis of cancer cells. Our results were in contrast to recent reports, which demonstrated that the downregulation of PNKY increased neuronal production without influencing NSC proliferation [9]. Additionally, our data indicate that PNKY exhibits a tumorigenic function. It can act as an onco-lncRNA in breast cancer by promoting cell cycle progression and cell proliferation and by restricting cell death in the MDA-MB-231 and MCF7 breast cancer cells. 

Recent reports demonstrated that epithelial–mesenchymal transition is an essential process in cancer cell metastasis and invasion [13]. The loss of E-cadherin and the upregulation of Vimentin are important events in EMT and cancer cell migration [14,15]. Our data indicate that PNKY was clearly upregulated in high-grade, poorly differentiated lymph node metastasis breast tumors. The migration of breast cancer cells was also restricted following PNKY downregulation. Furthermore, we found that siRNA-mediated PNKY suppression upregulates Vimentin, Snail, and Zeb-1, consequently repressing E-cadherin expression. In line with the above results, our findings demonstrated that miR-150, which represses Zeb1 [16,17,18], and the miR-29 family, which restricts EMT [19,20], were upregulated when PNKY was suppressed in the MDA-MB-231 cell line. Therefore, we conclude that PNKY may prompt the migration of breast cancer cells and contribute to breast cancer cell metastasis by affecting EMT. 

In this study, we demonstrated that the downregulation of PNKY triggers cellular senescence. We established that senescent cells increased following PNKY suppression. In breast cancer cells, increasing evidence indicates that cellular senescence, together with apoptosis, can inhibit tumor progression [21,22,23]. Therefore, based on these findings, PNKY could be used as a potential therapeutic target to prevent cancer progression by inducing cellular senescence and apoptosis in cancer cells, which irreversibly exits them from the cell cycle. 

In conclusion, we provide evidence that lncRNA PNKY is expressed in several cancer types, and specifically, it upregulates in breast tumor tissues. Our results are the first to demonstrate the biological role of PNKY in tumorigenesis, tumor progression, and metastasis of breast cancer. We therefore provide novel evidence about the function of PNKY in cancer cells, which can be considered as a novel biomarker with potential diagnostic, prognostic, and therapeutic value.

## 4. Materials and Methods

### 4.1. Clinical Sample Collection

In the current study, tumor tissue samples and corresponding noncancerous tissues were obtained from 47 patients with breast cancer from the Iran National Tumor Bank, which is founded by the Cancer Institute of the Tehran University of Medical Sciences, for cancer research (Tehran, Iran). The tissues had been immediately snap-frozen in liquid nitrogen and stored at −185 °C until they were ready for RNA extraction. The Ethics Committee of the Kerman Graduate University of Advanced Technology approved the experimental procedure. Prior to participation, the Iran National Tumor Bank obtained patients’ written informed consent. The clinicopathological data associated with each patient was obtained from the Iran National Tumor Bank (Table 1). Additionally, in order to investigate the potential expression profile of PNKY in colorectal, brain, and prostate cancers, we gathered 10 tissue samples for each cancer from the Iran National Tumor Bank.

### 4.2. Cell Culture

NCCIT (human embryonic carcinoma cell line), MCF7, MDA-MB-231, and SK-BR-3 (human breast cancer cell line) were obtained from the national cell bank of Iran (Pasteur Institute, Tehran, Iran). MCF7 cell lines were cultured in RPMI-1640 (Gibco, New York, NY, USA), enriched with 10% fetal bovine serum, 100 U/mL penicillin, and 10 μg/mL streptomycin. MDA-MB-23, NCCIT, and primary hDPSCs (human dental pulp stem cells) were cultured in high glucose DMEM (Gibco, New York, USA), supplemented with 15% fetal bovine serum, 100 U/mL penicillin, and 10 μg/mL streptomycin.

### 4.3. RNA Extraction, cDNA Synthesis, and Quantitative Real-Time PCR 

Total RNA from tissue samples and cultured cells was isolated using Trizol solution (Invitrogen, Waltham, MA, USA) according to the manufacturer’s instructions. The quantity of the extracted RNA was measured by UV spectrophotometry (Cary 60, Mulgrave, Australia) at 260 nm and its quality was proved by observing the RNA samples on 1% agarose gel electrophoresis. To remove the genomic DNA, RNase-free DNase I (Fermentase, Vilnius, Lithuania) treatment of total RNA was performed according to the manufacturer’s instructions. The first strand of cDNA was synthesized by using 1 µg RNA, 200 U/µL MMLV reverse transcriptase (Fermentase), 20U RNase inhibitor, dNTP mix (final concentration of 1 mM) with random hexamer priming in a 20 µL reaction. For each sample, a no-RT control was used in parallel to detect any potential contamination with genomic DNA. Specific primers were designed for PNKY, β-actin, and other genes by using Gene Runner software, version 4.0, while the Basic Local Alignment Search Tool (BLAST) was used to check the confidence of unity attachment of primers (Table 2). 

The qPCR reaction was done using SYBR Premix Ex TaqTM II (Takara, Shiga, Japan) on the Rotor-Gene 6000 instrument (Corbett Life Science, Sydney, Australia). Relative expression of specific RNAs was calculated using 2^−ΔΔCt^ method and it was normalized based on the expression level of β-actin as an internal control. 

### 4.4. Isolation of Cytoplasmic and Nuclear Fractionation

Subcellular fractionation was conducted to localize the PNKY transcript in the MDA-MB-231 cells, as described previously [24]. Briefly, the cells were harvested and the pellet was re-suspended in 2 mL PBS, 2 mL nuclear isolation buffer (1.28 M sucrose; 40 mM Tris-HCl pH 7.5; 20 mM MgCl2; 4% Triton X-100), and 6 mL water on ice for 20 min. Centrifuge at 2500× *g* for 15 min was performed to isolate the nuclei of cells. Then, total RNA from both cytoplasm and nucleus was extracted for PNKY and U6 expression analysis.

### 4.5. Neural Differentiation of NCCIT Cells 

The neural differentiation procedure was done as described previously with some modification [25]. In brief, NCCIT cells were plated on a T75 flask, and differentiation was induced by adding 10 μM trans-RA to the medium every two days for 14 days. After that, RA was removed from the media, and the cells were then harvested at different time points (3rd, 7th, 11st and 14th day). We further examined the accuracy of the differentiation process by assessing the expression level of stemness related genes (Oct4, Nanog, Nestin and Nucleostemin) and differentiation specific gene (MBP). 

### 4.6. PNKY Silencing by siRNA Transfection 

For silencing PNKY, two specific siRNAs directed against PNKY were designed by using the siRNA selection program (Table 3) (Whitehead Institute for Biomedical Research, http://jura.wi.mit.edu/, accessed on 27 October 2022) and scrambled siRNA with no known RNA target in the cells was also purchased from Dharmacon. MDA-MB-231 and MCF7 cells at a confluency of 40–60% were transfected with 100 nmol/mL of PNKY-siRNA and scrambled siRNA using lipofectamine 2000 (Invitrogen) according to the manufacturer’s instructions. Then, the cells were harvested for genes quantification studies to determine the efficiency of PNKY silencing.

### 4.7. Colony Formation Assay

The effect of suppression of PNKY on clonogenic ability of breast cancer cells was analyzed by colony formation assay. One day after cell transfection with siRNAs, the limited fraction of transfected cells was seeded in each well of plate in a media containing 10% FBS. After 2 weeks, the fixed colonies were stained with 0.1% crystal violet (Sigma, St. Louis, MO, USA) in PBS for 15 min. After this process, visible colonies were counted under stereomicroscopy and analyzed by ImageJ program (NIH, Bethesda, MD, USA).

### 4.8. Cell Senescence-Associated Beta-Galactosidase Assay

The senescent cells are recognized by detecting an increased level of lysosomal β-galactosidase activity. Briefly, the cells fixed by paraformaldehyde 4% and glutaraldehyde 0.2%, and then stained by X-gal (5-Bromo-4-chloro-3-indolyl β-D-galactopyranoside) staining solution (1 mg/mL x-gal, 40 mM citric acid/sodium phosphate, 0.15 M NaCl, 2 mM MgCl_2_, 5 mM potassium ferrocyanide, and 5 mM potassium ferricyanide) at pH 5.9–6.1. Finally, we quantified the percentage of SA-β-gal-positive stained cells by counting the number of positive stained cells in randomly selected fields (n = 3) under microscopy. 

### 4.9. Flow Cytometric Analysis

For cell cycle analysis, two days after siRNAs transfection, the transfected cells were harvested by treating them with Trypsin-EDTA and washing in PBS. To prepared single cells, the harvested pelleted cells were re-suspended in PBS. To stain the cells, the propidium iodide staining solution containing 50 μg/mL propidium iodide, 0.1% Triton X-100 and 0.1 % sodium citrate was then added to the cells. To end this, the stained singular cells were finally analyzed employing flow cytometric instrument (Partec, Jettingen-Scheppach, Germany) [26]. Cell cycle profiles were analyzed using flowjo 7.6.1 software. To measure the apoptosis rate, a fraction of the collected singular cells was marked with FITC-Annexin V and propidium iodide (PI) according to the manufacturer’s protocol. Then, the cells were analyzed on a flow cytometry and the proportion of apoptotic cells versus live cells was determined. 

### 4.10. Cell Proliferation Assay

MDA-MB-231 and MCF7 cells were seeded into 96-well plates and they were stained with 100 µL sterile MTT dye (0.5 mg/mL, Sigma) at specified time points. After discarding media from the cell cultures, 150 µL of dimethyl sulfoxide (DMSO, Sigma) was added to the cells. The absorbance of lysate cells was measured at 570 nm. 

### 4.11. Migration Assay

The influence of siRNA-mediated PNKY downregulation on cell migration was investigated by using wound healing assay. The cells were seeded on the 12-well plates and cultured until 60% confluency. At this confluency, the cancer cells were transfected with siRNAs and 12–18 h post transfection, they were starved in the culture medium containing 1% FBS. A straight scratch, simulating that a wound has been made into a confluent monolayer cell by using a sterile white pipette tip. Then, we measured and photographed the cell migration inward to fill the void at specified time points. 

### 4.12. Western Blotting

The cells were gathered and lysed with lysis buffer (50 mM Tris, 0.15 M NaCl, 1 mM EGTA, 1% NP40, 0.25% SDS, pH 7.4). The isolated proteins were separated on 12% SDS-polyacrylamide gel, then they were transferred onto a polyvinylidene fluoride membrane (PVDF; Roche). The membrane was blocked for 1 h at 4 °C by floating in a blocking buffer (5% BSA in Tris-buffered saline containing 0.1% Tween-20). The membrane was then incubated with appropriate dilution of primary antibody Cyclin D1, P16, Cox2, GAPDH, β-Actin (Abcam, Waltham, MA, USA) in blocking buffer at 4 °C overnight. Then, horseradish peroxidase (HRP)-conjugated secondary antibody added to the membrane at room temperature for 1 h. Complex primary antibody and target protein was detected using an enhanced chemiluminescence detection kit (ECL Amersham, GE Healthcare, Danderyd, Sweden). We employed Image Lab3 analyzing software (Bio-Rad, Hercules, CA, USA) to determine the intensity quantification of the protein bands.

### 4.13. MicroRNA Expression Analysis

Total RNA isolation, cDNA synthesis, and the qPCR of microRNAs were performed using the BonmiR kit (BON209002, Bon Yakhteh Company, Tehran, Iran), according to manufacture instructions. Briefly, miRNA was extended in a polyadenylation reaction, and then the elongated miRNA was used as a template to produce cDNA. The real-time PCR was done using the BonmiR kit and specific miRNA primer on Rotor-Gene 6000 instrument (Corbett Life Science, Oatley, Australia). 

### 4.14. Statistical Analysis

All experiments were replicated three times. We analyzed the relative level of gene expression using 2^−ΔΔCt^ method. The statistical difference between groups was determined by the one-way ANOVA and independent student’s *t*-test that was performed by the GraphPad prism 6.07 software (GraphPad Software, San Diego, CA, USA) and the REST 2009 program. The *p* value of less than 0.05 was considered significant. 

## Figures and Tables

**Figure 1 ncrna-09-00025-f001:**
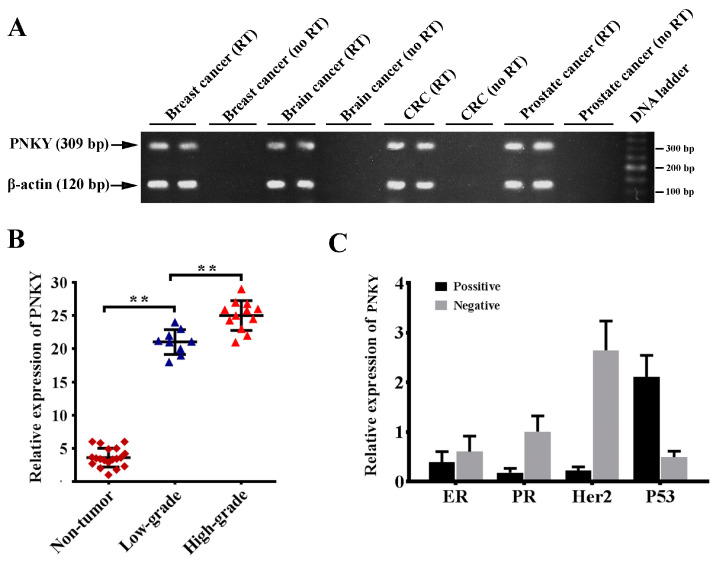
PNKY is upregulated in breast cancer. (**A**) PNKY transcript was detected in several tumor tissue types, including breast, brain, prostate, and colorectal cancer tissues using RT-PCR. Because PNKY has one exon, a no-RT control was performed for each sample in parallel to detect any potential contamination with genomic DNA. (**B**) PNKY is significantly overexpressed in breast tumor samples compared with non-tumoral breast tissues; therefore, its expression is significantly upregulated in high-grade breast tumors. (**C**) Bar charts indicate that PNKY is upregulated in ER, PR, Her2 negative, and P53 positive tumor samples. The values shown represent the mean ± SE. ** *p* value < 0.01.

**Figure 2 ncrna-09-00025-f002:**
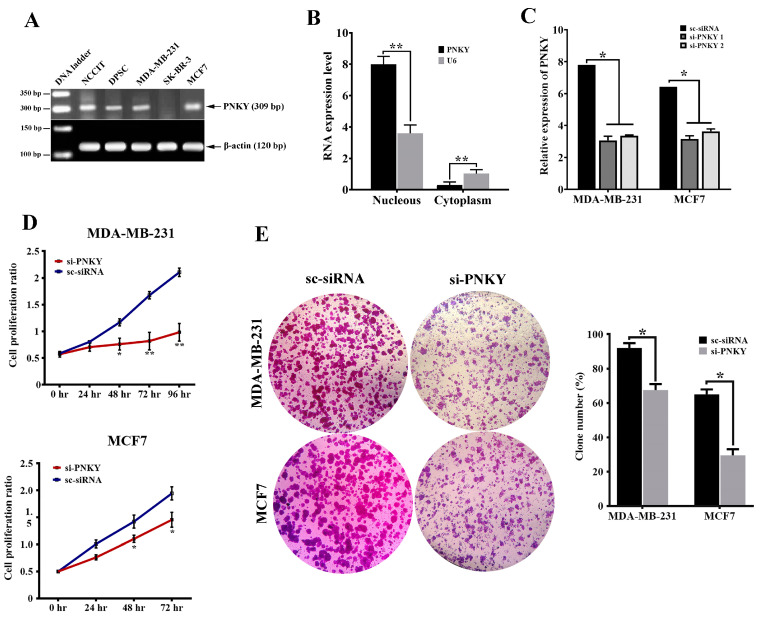
PNKY is expressed in stem/cancer cells and promotes cancer cells proliferation. (**A**) The expression profile of PNKY in cancer/stem cells showed that it is expressed in NCCIT, DPSC, MDA-MB-231, and MCF7 cells. (**B**) The PNKY transcript is localized in the nucleus of breast cancer cells. (**C**) PNKY was efficiently knocked down in MCF7 and MDA-MB-231 cell lines after transfection with specific siRNAs against the PNKY transcript. (**D**) A cell proliferation assay indicated that the growth of MCF7 and MDA-MB-231 cells significantly decreased when PNKY was silenced. (**E**) Colony formation ability in PNKY suppressed cells decreased compared with transfected cells by scrambled siRNA. The values shown represent the mean ± SE. * *p* value < 0.05, ** *p* value < 0.01.

**Figure 3 ncrna-09-00025-f003:**
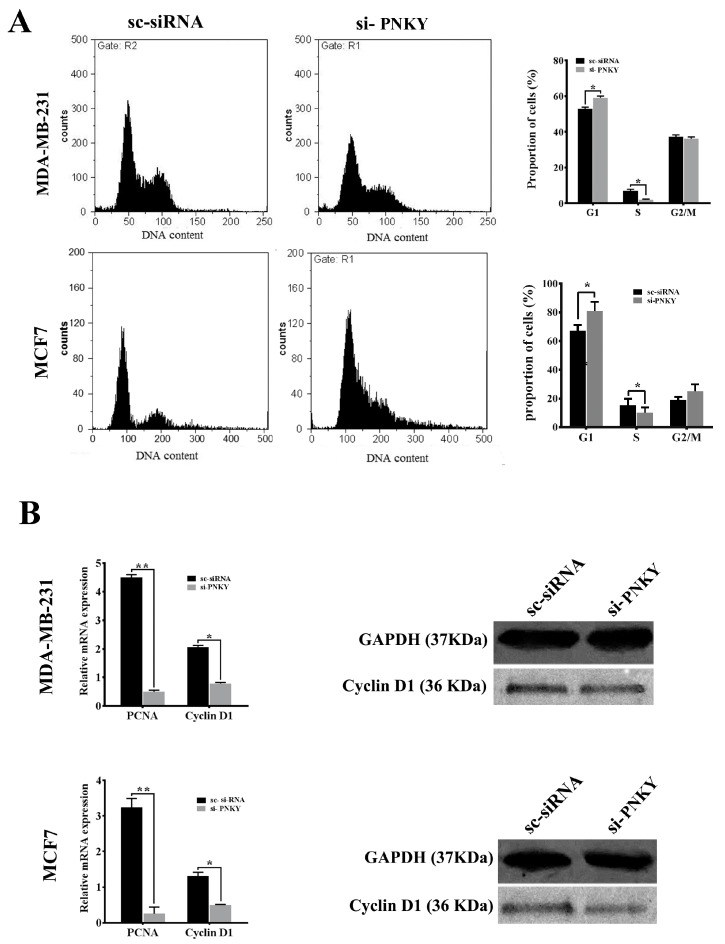
Knockdown of PNKY induces cell cycle arrest in MCF7 and MDA-MB-231 cell lines. (**A**) Cell cycle analysis of MDA-MB-231 and MCF7 cells indicates that the proportion of cells in the G1 phase of the cell cycle increased when PNKY was suppressed. (**B**) RT-qPCR analysis of PCNA and Cyclin D1 revealed that these transcripts significantly downregulate after PNKY is knocked down. In addition, the western blotting analysis showed that the expression level of Cyclin D1 significantly decreased following the downregulation of PNKY. The values shown represent the mean ± SE. * *p* value < 0.05, ** *p* value < 0.01.

**Figure 4 ncrna-09-00025-f004:**
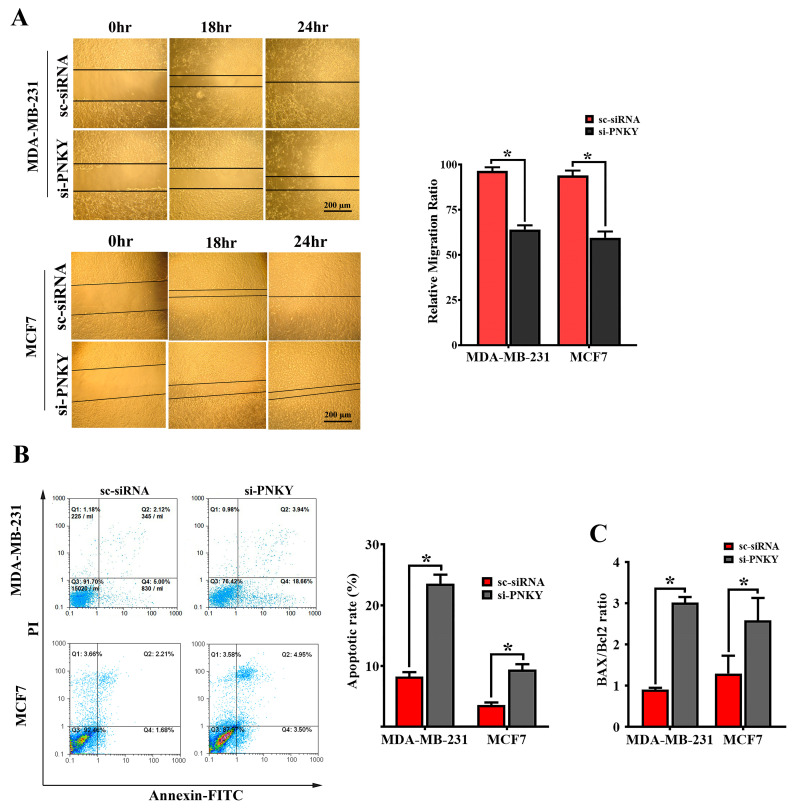
PNKY restricts apoptosis and promotes the migration of MCF7 and MDA-MB-231 cells. (**A**) Silencing PNKY in MCF7 and MDA-MB-231 cells restricts cell migration. (**B**) Annexin V/PI (propidium iodide) staining demonstrated that the downregulation of PNKY in MCF7 and MDA-MB-231 cells induces apoptosis. (**C**) The expression of Bcl2/Bax, a well-defined apoptosis marker, was altered following PNKY suppression. The values shown represent the mean ± SE. * *p* value < 0.05.

**Figure 5 ncrna-09-00025-f005:**
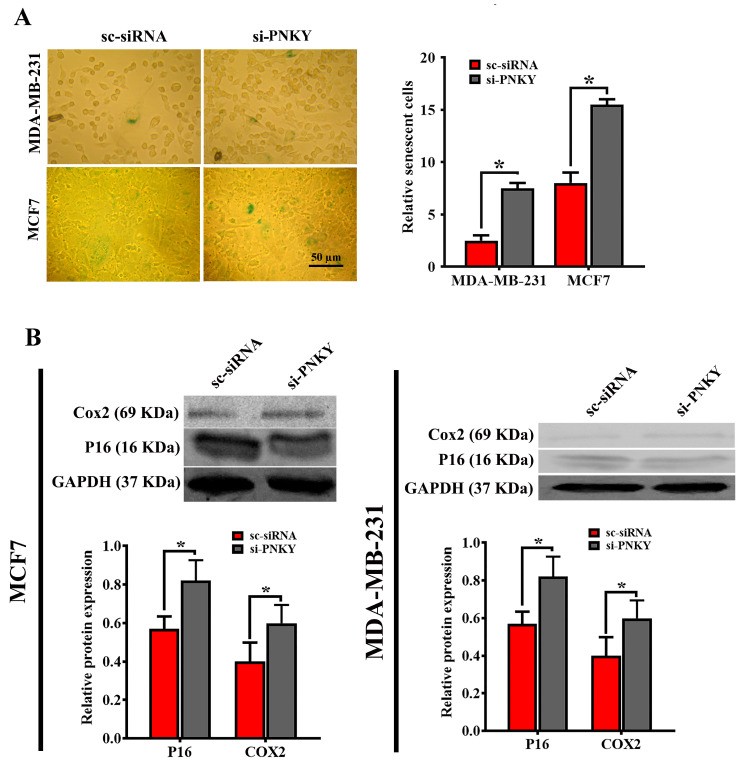
The downregulation of PNKY promotes cellular senescence in MCF7 and MDA-MB-231 cells. (**A**) The number of SA-β-gal positive staining breast cancer cells significantly increased following PNKY downregulation. (**B**) Western blotting analysis showed that P16 and COX2, two well-known senescence markers, were significantly upregulated in breast cancer cells when PNKY was suppressed. The values shown represent the mean ± SE. * *p* value < 0.05.

**Figure 6 ncrna-09-00025-f006:**
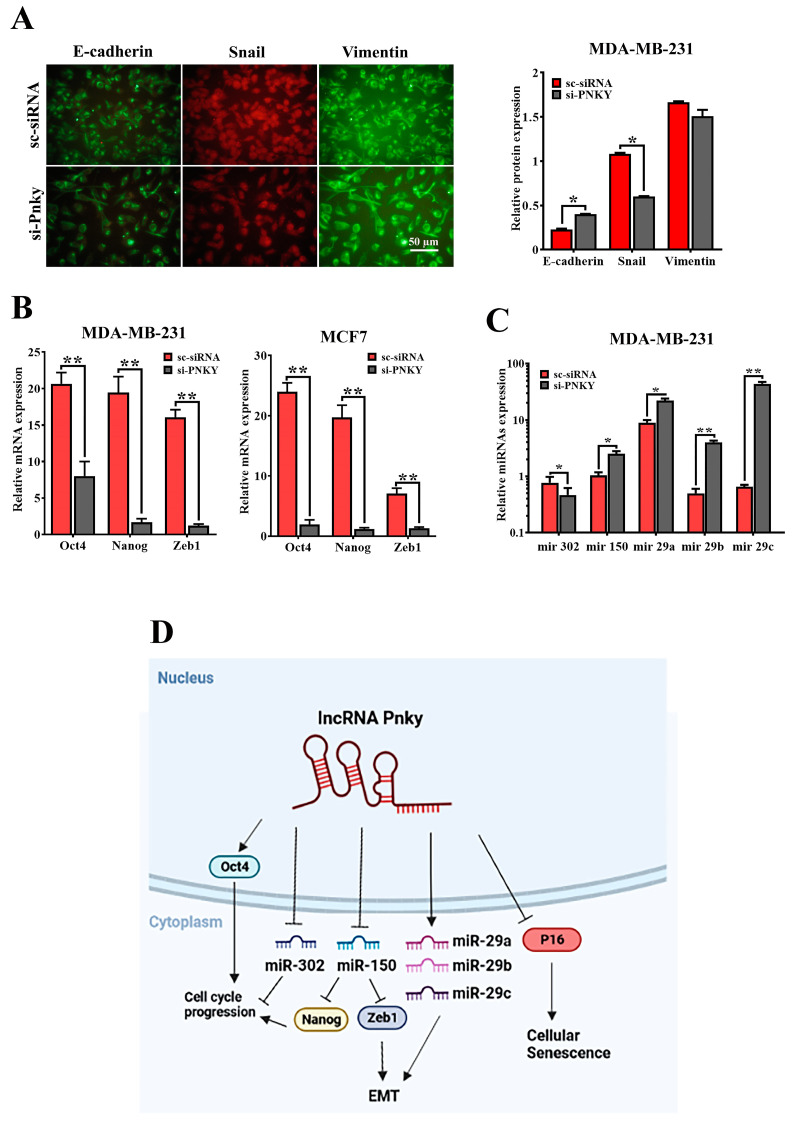
PNKY promotes EMT in breast cancer cells. (**A**) Immunofluorescence staining of EMT markers revealed that PNKY suppression upregulates epithelial marker (E-Cadherin) and downregulates mesenchymal markers (Snail and Vimentin) in MDA-MB-231 cells (**B**) Oct4/Nanog, stemness factors, and Zeb1, a well-defined EMT marker, were significantly downregulated following PNKY suppression. (**C**) The expression levels of hsa-miR-302, hsa-miR-150, hsa-mir-29a, hsa-mir-29b, and hsa-mir-29c were altered following PNKY silencing in the MDA-MB-231 cell line. (**D**) Schematic diagram illustrates the suggested PNKY function in breast cancer. The values shown represent the mean ± SE. * *p* value < 0.05, ** *p* value < 0.01.

**Table 1 ncrna-09-00025-t001:** Co-relationship between PNKY expression level and clinicopathologic characteristics of breast cancer tissues.

Clinicopathologic Parameters	Number of Cases	*p* Value
Tumor samples Normal samples	47 47	0.004 *
**Age (years)**		0.76
≤65	35	
>65	12	
**Tumor size**		0.63
≤5	35	
>5	12	
**Histologic grade**		0.008 **
Low grade	26	
High grade	21	
**ER status**		0.1
Negative	13	
Positive	34	
**PR status**		0.7
Negative	19	
Positive	28	
**HER2 status**		0.08
Negative	32	
Positive	15	
**TNM stage**		0.6
I, II	19	
III, IV	28	
**P53 status**		0.17
Negative	27	
Positive	12	
**Lymphatic node invasion**		
Negative	18	0.011 *
Positive	29	
**Histology**		0.07
Lobular	7	
Ductal	40	

* *p* < 0.05 ** *p* < 0.01.

**Table 2 ncrna-09-00025-t002:** Primer sequences.

Gene	Primer (Forward) 5′–3′	Primer (Reverse) 5′–3′	Amplicon Size
PNKY	AGCTCTCGCTGGTTTTAGG	GTGAGGGAGATATCAAGACACC	309 bp
PNKY	AAGCACGTTGAAGGTGTCTC	CATTGTCCTAGCGAGTGATC	214 bp
β-actin	ACCACCTTCAACTCCATCATG	CTCCTTCTGCATCCTGTCG	120 bp
U6	GCTTCGGCAGCACATATACTAAAAT	CGCTTCACGAATTTGCGTGTCAT	89 bp
Nestin	GAGAAACAGGGCCTACAGAG	GCTGAGGGAAGTCTTGGAG	168 bp
Nucleostemin	GGGAAGATAACCAAGCGTGTG	CCTCCAAGAAGTTTCCAAAGG	98 bp
Oct4(POU5F1)	AGTGAGAGGCAACCTGGAGA	TTACAGAACCACACTCGGACC	140 bp
MBP	CACATGTACAAGGACTCACAC	GAAGAAGTGGACTACTGGGT	108 bp
Nanog	TAACCTTGGCTGCCGTCTCT	AAGCAAAGCCTCCCAATCC	154 bp
Bax	GGACGAACTGGACAGTAACATGG	GCAAAGTAGAAAAGGGCGACAAC	150 bp
Bcl2	CTGCACCTGACGCCCTTCACC	CACATGACCCCACCGAACTCAAAGA	119 bp
Cyclin D1	ACAAACAGATCATCCGCAAACAC	TGTTGGGGCTCCTCAGGTTC	144 bp
PCNA	AGGTGGAGAACTTGGAAATGG	CGTTGAAGAGAGTGGAGTGG	160 bp
Vimentin	CTCTTCCAAACTTTTCCTCCC	AGTTTCGTTGATAACCTGTCC	134 bp
Zeb1	ACCCTTGAAAGTGATCCAGC	CATTCCATTTTCTGTCTTCCGC	142 bp

**Table 3 ncrna-09-00025-t003:** siRNAs sequence.

Name	Sequence	
siRNA 1	Sense	5′ACACAAACCUCCCAAAUdtdt3′
	Antisence	5′UGUGUUUGGAGGGUUUAdtdt3′
siRNA 2	Sense	5′CAAAUUUCCACCUGAGUAAdtdt3′
	Antisence	5′UUACUCAGGUGGAAAUUUGdtdt3′

## Data Availability

Data generated at the central facility is available upon request.

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
