# Peer review of "LncRNA PNKY Is Upregulated in Breast Cancer and Promotes Cell Proliferation and EMT in Breast Cancer Cells"

_ncrna, 2023, doi:10.3390/ncrna9020025_

Round 1

Reviewer 1 Report

In the article by Hakiminia et al, authors investigate the role of the lncRNA PNKY in breast cancer progression. The phenotypes observed upon PNKY knockdown are significant and most of the results are clear.

However, I have the following major criticisms regarding the manuscript

1.       Authors repeatedly talk about cancer stem cells in the manuscript and have put an effort to connect the previously characterized role of this lncRNA in neuronal stem cell differentiation and cancer stem cells. Even the abstract has approx. 20% of this concept. This is unwanted and not really sensible. This manuscript hardly deals with CSCs and there is no data presented in that direction. Max 2-3 sentences on potential CSC vs neuronal stem cell role of PNKY should be limited to the discussion section.

2.       In Figure 1A, authors state that data is presented from 10 samples each from four cancer types. The data presented is only for two samples each with minus RT for only one as I can understand. The labeling on these gels are also not clear. If authors want to state that they have looked at specific detection of PNKY in 40 samples, +/-RT data for all needs to be shown. This can also be a supplementary figure as the full set of quantitative data is focused on breast cancer only

3.       Line 75: “We further explored….ductal breast tumor tissues”- where is this data?

4.       Figure 1C and line 76 to 82: The authors state that the p53 status dependence data is not significant. What about the other statements made here? No significance labeling or p-values are stated in the figure/legends. If the data doesn’t show statistical significance, how are these conclusions made?

5.       Figure 1E- Y-axis label (PNKY/U6RNA ratios) is misleading. Please remove gene names from the axis label

6.       I do not see any significance or connection of Figure 2A/B data on neuronal differentiation to the main story in this manuscript. It is unnecessary and this data doesn’t add anything new to the story-line. Integrating the stem-cell transcription factors information from neuronal system and discussing in the breast cancer side is not acceptable. I would suggest Figure 1D/E to be moved as Figure 2A/B and maintain the flow of the manuscript. In that way, Figure 2 will be completely on the breast cancer cells lines- expression verification, subcellular localization, knockdown (proliferation, colony formation). The neuronal differentiation data can be maximum only a disconnected supplementary figure which cannot have much impact on abstract and text.

7.       Transient siRNA transfections are usually sustainable for only 48-96h max and it is unusual to do colony formation assay with siRNA transfected cells. Are these 2 week long assays done after transient transfection?  The siRNA transfection protocol (especially time-frame) and how it is integrated to the phenotypic analyses is pending from methods (e.g. 24h post-transfection, cells were seeded for MTT…. What is time 0?)

8.       Many of the statements are superlative and the language needs to be proofed. For example, line number 106 “…expression of PNKY was enormously suppressed….”. Enormous is too much for 75% knockdown.

9.       Line 122-123” ….the identified genes contributed in cell proliferation………..”, the meaning is not clear

10.   Line 351-352:” …cells were harvested in an appropriate manner and…..” is vague and not acceptable. Again, no information is provided regarding how long after siRNA treatment is the analyses done. Moreover, mentioning reduced S phase population, consistent with less PCNA is more relevant than discussing compensatory increase in G1 population.

11.   Line 136-137: I don’t think scratch wound healing assays are quantitative enough to measure the speed of migration. Please remove “speed” from here and in methods. Despite reduction in FBS levels, wound healing assays may not be a direct readout for migration. Since the PNKY siRNA significantly affect viability and proliferation, the potential effect of proliferation/death on this result should be discussed or assays should be done in the presence of mitomycin-C. Again the details of the methods are missing (how many hours after siRNA transfection/medium change, migration assays were performed etc.)

12.   Figure 4C: It is hard to say what data is presented here? How can the realtime data for BCL2 and BAX shown in same bar graphs?

13.   Most of the western blots are overexposed and saturated and this is really bad in case of p16 blot in Fig 5B. p16 is a marker for senescence and how can one explain suppression of p16 in siPNKY cells, when actually they show enhances SA-Bgal staining? This is wrongly stated in line 256 of discussion. Also, how is the senescence phenotype assessed? Do you mean that transient siRNA transfected cells are passaged? Or different passage cells are transfected with siRNA? How long is the siRNA treatment done before SA-Bgal staining? This senescence data lacks methods and is weak. My suggestion is to avoid this and retain conclusive findings with clear methodology and evidence (which the manuscript certainly have).

14.   Unlike the other experiments, the data in Fig. 6C figure panel on miRNA expression is provided only for a single cell line and no information on the cell line used is provided. This should be stated clearly in the text and also in the figure/figure legends. Also in the discussion line 251: it can be clearly stated that “ ….when PNKY was silenced in xxx cell line”.

15.   Generally avoid the terminology of “PNKY inhibition” for siRNA treatment and use d”down-regulation” or “suppression” consistently

16.   Line 258: “ we concluded the expression of PNKY was dramatically decreased in senescent cells……..”. How is this conclusion made?

17.   The discussion is rather redundant with results and can be cut short or edited.

18.   Table 1 needs to be cited in the results section.

Author Response

-           We thank the referee for her/his insightful comments on our study.

In the article by Hakiminia et al, authors investigate the role of the lncRNA PNKY in breast cancer progression. The phenotypes observed upon PNKY knockdown are significant and most of the results are clear.

However, I have the following major criticisms regarding the manuscript

  1. Authors repeatedly talk about cancer stem cells in the manuscript and have put an effort to connect the previously characterized role of this lncRNA in neuronal stem cell differentiation and cancer stem cells. Even the abstract has approx. 20% of this concept. This is unwanted and not really sensible. This manuscript hardly deals with CSCs and there is no data presented in that direction. Max 2-3 sentences on potential CSC vs neuronal stem cell role of PNKY should be limited to the discussion section.

Response: we agree with you and now we edited the abstract and introduction.

  1. In Figure 1A, authors state that data is presented from 10 samples each from four cancer types. The data presented is only for two samples each with minus RT for only one as I can understand. The labeling on these gels are also not clear. If authors want to state that they have looked at specific detection of PNKY in 40 samples, +/-RT data for all needs to be shown. This can also be a supplementary figure as the full set of quantitative data is focused on breast cancer only.

Response: we made a pool of RNA from 10 samples for each cancer types. The labeling of DNA laddering is now edited in the figure.

  1. Line 75: “We further explored….ductal breast tumor tissues”- where is this data?

Response: This data has been presented in Table 1 (p=0.07) and seems significant broadly, so we didn’t show it as diagram and now we refereed in the main text.

  1. Figure 1C and line 76 to 82: The authors state that the p53 status dependence data is not significant. What about the other statements made here? No significance labeling or p-values are stated in the figure/legends. If the data doesn’t show statistical significance, how are these conclusions made?

Response: These data are summarized in Table 1 and the p value is now added in the text.

  1. Figure 1E- Y-axis label (PNKY/U6RNA ratios) is misleading. Please remove gene names from the axis label.

Response: we agree with you and now revised it.

  1. I do not see any significance or connection of Figure 2A/B data on neuronal differentiation to the main story in this manuscript. It is unnecessary and this data doesn’t add anything new to the story-line. Integrating the stem-cell transcription factors information from neuronal system and discussing in the breast cancer side is not acceptable. I would suggest Figure 1D/E to be moved as Figure 2A/B and maintain the flow of the manuscript. In that way, Figure 2 will be completely on the breast cancer cells lines- expression verification, subcellular localization, knockdown (proliferation, colony formation). The neuronal differentiation data can be maximum only a disconnected supplementary figure which cannot have much impact on abstract and text.

Response: we thank for your insightful suggestion and now we revised the figures.

  1. Transient siRNA transfections are usually sustainable for only 48-96h max and it is unusual to do colony formation assay with siRNA transfected cells. Are these 2 week long assays done after transient transfection? The siRNA transfection protocol (especially time-frame) and how it is integrated to the phenotypic analyses is pending from methods (e.g. 24h post-transfection, cells were seeded for MTT…. What is time 0?)

Response: we revised the method section related to colony formation assay.

  1. Many of the statements are superlative and the language needs to be proofed. For example, line number 106 “…expression of PNKY was enormously suppressed….”. Enormous is too much for 75% knockdown.

Response: we agree with you and now edited in the text.

  1. Line 122-123” ….the identified genes contributed in cell proliferation………..”, the meaning is not clear

Response: we agree with you and now we revised it.

  1. Line 351-352:” …cells were harvested in an appropriate manner and…..” is vague and not acceptable. Again, no information is provided regarding how long after siRNA treatment is the analyses done. Moreover, mentioning reduced S phase population, consistent with less PCNA is more relevant than discussing compensatory increase in G1 population.

Response: we analyzed cellular assays two days after transfection and now we edited it in the methods section. In addition, we added the relation of downregulation of PCNA with decreasing S Phase population in discussion section.

  1. Line 136-137: I don’t think scratch wound healing assays are quantitative enough to measure the speed of migration. Please remove “speed” from here and in methods. Despite reduction in FBS levels, wound healing assays may not be a direct readout for migration. Since the PNKY siRNA significantly affect viability and proliferation, the potential effect of proliferation/death on this result should be discussed or assays should be done in the presence of mitomycin-C. Again the details of the methods are missing (how many hours after siRNA transfection/medium change, migration assays were performed etc.)

Response: we agree with you and now we deleted speed from migration section and method section was also revised by adding time details.

  1. Figure 4C: It is hard to say what data is presented here? How can the realtime data for BCL2 and BAX shown in same bar graphs?

Response: we analyzed the ratio expression of BCL2 to BAX.

  1. Most of the western blots are overexposed and saturated and this is really bad in case of p16 blot in Fig 5B. p16 is a marker for senescence and how can one explain suppression of p16 in siPNKY cells, when actually they show enhances SA-Bgal staining? This is wrongly stated in line 256 of discussion. Also, how is the senescence phenotype assessed? Do you mean that transient siRNA transfected cells are passaged? Or different passage cells are transfected with siRNA? How long is the siRNA treatment done before SA-Bgal staining? This senescence data lacks methods and is weak. My suggestion is to avoid this and retain conclusive findings with clear methodology and evidence (which the manuscript certainly have).

Response: Thanks for your insightful comment and we analyzed western blotting again and we found that the analysis was mistaken and now we revised this mistake.

  1. Unlike the other experiments, the data in Fig. 6C figure panel on miRNA expression is provided only for a single cell line and no information on the cell line used is provided. This should be stated clearly in the text and also in the figure/figure legends. Also in the discussion line 251: it can be clearly stated that “ ….when PNKY was silenced in xxx cell line”.

Response: we agree with you and now edited it in the text and legends.

  1. Generally avoid the terminology of “PNKY inhibition” for siRNA treatment and use d”down-regulation” or “suppression” consistently

Response: we agree with you and now revised it in MS.

  1. Line 258: “ we concluded the expression of PNKY was dramatically decreased in The discussion is rather redundant with results and can be cut short or edited.

Response: we agree with you and now edited it.

  1. Table 1 needs to be cited in the results section.

Response: we agree and now we edited the manuscript. The modified sections were highlighted in revised MS.

  1. senescent cells……..”. How is this conclusion made?

Response: we revised it in the text. In independent study by another my student we found that PNKY expression related to senescence in stem cells (dental pulp stem cells) and this data yet unpublished.

Reviewer 2 Report

Hakiminia et al in this manuscript found PNKY was upregulated in breast tumors. Knock down of PNKY restricts BCC proliferation. PNKY might be triggering EMT through upregulated mir-150 and restricting Zeb1 and snail.

Generally speaking, the data supports the analysis for the most part. However, further experiments and classification should be included to make the results more convincible.

Comments

1, Some English syntax and typos need to be addressed.

2, The authors should add the size (base pairs) for the DNA ladder for figure 1A and 1D.

3, The authors should classify how they define the high- and low-grade breast tumors, and the result from table 1 shows that 9/10 low grade patients have high level of PNKY, and 12/21 high grade patients have high level of PNKY, it is hard to get the conclusion “the significant upregulation of PNKY in the high- grade ductal breast tumors versus low- grade ones”.

4, The authors should discuss the result about “PNKY was downregulated in lobular breast tumor samples compared with ductal breast tumor tissues.”

5, Fig 1E, the label si-PNKY should be PNKY, and all the calculation for the qRT-PCR is not clear, it is not based on the 2-ΔΔCt method.

6, Are there any statistical differences for cell proliferation in Fig 2D?

7, It is hard to see the inhibition of cyclin D1 protein by si-PNKY in Fig 3B, the author can’t get a conclusion “dramatically downregulated when PNKY was silenced (Fig 3B).” The authors should add the description in Methods of how they quantify the western blot results.

8, All the microscope pictures should include scar bar.

9, It is better to use Western Blot for verification of EMTs protein level in Fig 6A.

10, The big problem of this manuscript is lack of rational reason for the study, for example, the authors did not tell the reader why they detect the miRNAs, and they did not use the classical model to verify the stem cell phenotype etc.

Author Response

- We thank the referee for her/his insightful comments on our study.           .

  1. Hakiminia et al in this manuscript found PNKY was upregulated in breast tumors. Knock down of PNKY restricts BCC proliferation. PNKY might be triggering EMT through upregulated mir-150 and restricting Zeb1 and snail.

Generally speaking, the data supports the analysis for the most part. However, further experiments and classification should be included to make the results more convincible.

Comments

1, Some English syntax and typos need to be addressed.

Response: we agree with you and now we edited the MS.

2, The authors should add the size (base pairs) for the DNA ladder for figure 1A and 1D.

Response: we agree with you and now we added the size for the DNA ladder.

3, The authors should classify how they define the high- and low-grade breast tumors, and the result from table 1 shows that 9/10 low grade patients have high level of PNKY, and 12/21 high grade patients have high level of PNKY, it is hard to get the conclusion “the significant upregulation of PNKY in the high- grade ductal breast tumors versus low- grade ones”.

Response: we obtained the tumor tissues from Iran national tumor bank and pathologist at this center determined the tumors grades. We submitted the MS for conference when our data was incomplete and the table is mis-uploaded during submission and now we revised it.

4, The authors should discuss the result about “PNKY was downregulated in lobular breast tumor samples compared with ductal breast tumor tissues.”

Response: we now added this section to discussion.

5, Fig 1E, the label si-PNKY should be PNKY, and all the calculation for the qRT-PCR is not clear, it is not based on the 2-ΔΔCt method.

Response: we agree with you and now we revised the fig 1E. we analyzed the qPCR by REST 2009 software which is based on the 2-ΔΔCt method.

6, Are there any statistical differences for cell proliferation in Fig 2D?

Response: the statistical analysis is now added to the figure.

7, It is hard to see the inhibition of cyclin D1 protein by si-PNKY in Fig 3B, the author can’t get a conclusion “dramatically downregulated when PNKY was silenced (Fig 3B).” The authors should add the description in Methods of how they quantify the western blot results.

Response: we agree with you and dramatically was removed and significantly was added in the result. We employed Image Lab3 analyzing software (Bio-Rad-USA) to determine the intensity quantification of protein bands and now added to methods section.

8, All the microscope pictures should include scar bar.

Response: we agree with you and now the scale bar added.

9, It is better to use Western Blot for verification of EMTs protein level in Fig 6A.

Response: Thanks for your suggestion but we have problem to prepare the EMT antibodies because grant limitation and purchasing them is time-consuming for us.

10, The big problem of this manuscript is lack of rational reason for the study, for example, the authors did not tell the reader why they detect the miRNAs, and they did not use the classical model to verify the stem cell phenotype etc.

Response: we agree with you and now we explain why detect the miRNAs. Thanks for your suggestion, in independent research by another my student we study the role of Pnky in dental pulp stem cells and to avoid the disrupt its integrity, we can present the data here.

Reviewer 3 Report

In the current study authors have shown role of lncRNA PNKY in breast cancer cells with some interesting findings, however, I have few suggestions for the authors which can be find below

1) Figure 1 (A and D) and other primer-based PCR results: this lncRNA has an overlapping transcript in UCSC Genome Browser (Gencode Transcript: ENST00000635423.1). Please specify the primer binding region in PNKY transcript and that primers used are specific for PNKY and doesn't bind the overlapping transcript.

2-Figure 1E: data presented here is very confusing. Why si-PNKY was used to confirm localization of this lncRNA? How they interpreted data about nuclear localization while using an siRNA? 

3-Figure 2A and all other images in manuscript: please add a scale bar to each image

4-Figure 3. western analysis: please quantify the images to support the conclusion presented in the text (can be done in triplicates for statistical analysis).

5-Figure 4.  (A) cell migration-Upper panel (middle image-18hours): Cells are clearly moving within the scar margin. Please adjust and repeat quantification.

6-Figure 5.  (B) Western blotting analysis in MDA-MB231 Panel: bands are too faint to quantify and interpret.

7- It is mentioned that two siRNA was used for PNKY, but it's not shown the siRNA efficiency of both siRNA was same or not. It should be mentioned that the results are from which siRNA. Are the results being reproducible with each siRNA?

Author Response

- We thank the referee for her/his insightful comments on our study.          

in the current study authors have shown role of lncRNA PNKY in breast cancer cells with some interesting findings, however, I have few suggestions for the authors which can be find below

  • Figure 1 (A and D) and other primer-based PCR results: this lncRNA has an overlapping transcript in UCSC Genome Browser (Gencode Transcript: ENST00000635423.1). Please specify the primer binding region in PNKY transcript and that primers used are specific for PNKY and doesn't bind the overlapping transcript.

Response: exon 2 of ENST00000635423.1 is overlap with Pnky but we designed the unique forward primer in unmatched region (300 nt from 5’     3’) which located from 239 to 258 nt.

2-Figure 1E: data presented here is very confusing. Why si-PNKY was used to confirm localization of this lncRNA? How they interpreted data about nuclear localization while using an siRNA?

Response: we agree with you and now we revised it in the MS.

3-Figure 2A and all other images in manuscript: please add a scale bar to each image

Response: we agree with you and now added scale bar.

4-Figure 3. western analysis: please quantify the images to support the conclusion presented in the text (can be done in triplicates for statistical analysis).

Response: we agree with you and now the analysis diagrams were revised in MS.

5-Figure 4.  (A) cell migration-Upper panel (middle image-18hours): Cells are clearly moving within the scar margin. Please adjust and repeat quantification.

Response: we agree with you and now we adjust it in MS.

6-Figure 5.  (B) Western blotting analysis in MDA-MB231 Panel: bands are too faint to quantify and interpret.

Response: It seems the expression of p16 and cox2 may be low in the cells.

7- It is mentioned that two siRNA was used for PNKY, but it's not shown the siRNA efficiency of both siRNA was same or not. It should be mentioned that the results are from which siRNA. Are the results being reproducible with each siRNA?

Response: we used two siRNAs (table 3) the efficiency of both siRNA was almost same. So, we continued with the siRNA1. 

Round 2

Reviewer 2 Report

The authors have satisfactorily addressed my concerns.

Author Response

Reply to Reviewer no 2:

Thanks for your comment.

Reviewer 3 Report

Thank you for addressing all my comments. Please address the following as well.

1- Figure 1 A- negative control (-RT) has band in all samples. Is it a genomic contamination?

2-FIG-1A: Figure legends and relevant text-Please clearly mention which cell lines/tissues/patient samples are used for this RT-PCR (not just cancer type).

3-In nuclear fractionation and PNKY level figure; please use appropriate statistics test.

Author Response

Prof. Zander Wu
Editor, Non-coding RNA,

Dear Professor Zander Wu,

We are delighted to hear our manuscript (MS# ncrna-2029857) has a chance for publication in Non-coding RNA, and we thank the referees for offering very insightful comments. We have tried to address all the final changes recommended by the referees a in our revised MS, as it is summarized below:

Reply to Reviewer no 3:

-           We thank the referee for her/his insightful comments on our study.

  • Figure 1 A- negative control (-RT) has band in all samples. Is it a genomic contamination?

Response: mis-labeling has been occurred when we revised the Fig and the labels were mis-moved. Now we editing it.   

  1. 2-FIG-1A: Figure legends and relevant text-Please clearly mention which cell lines/tissues/patient samples are used for this RT-PCR (not just cancer type).

Response: to investigate the potential expression of PNKY in colorectal, brain, breast and prostate cancer tissues, we gathered ten cancer tissues from ten patients with each cancer types including ten patients with colorectal cancer, ten patients with brain cancers, ten patients with breast cancer and ten patients with prostate cancer. After RNA extraction from tumor tissues, the RNA from ten tissue samples of each cancer was pooled and finally the expression of PNKY in the RNA pools was investigated. Now we mention it in the relevant sections.

  1. 3-In nuclear fractionation and PNKY level figure; please use appropriate statistics test.

Response: we agree with you and now revised it.